# Examining holistic processing strategies in dogs and humans through gaze behavior

**Soon Young Park**[1,2,3]*, **Diederick C. Niehorster**[4,5], **Ludwig Huber**[1,2,3], **Zsófia Virányi**[1,2,3]

1 Comparative Cognition, Messerli Research Institute, University of Veterinary Medicine Vienna, Vienna, Austria, 2 University of Vienna, Vienna, Austria, 3 Medical University Vienna, Vienna, Austria, 4 Lund University Humanities Lab, Lund University, Lund, Sweden, 5 Department of Psychology, Lund University, Lund, Sweden

* soonyoungpark01@gmail.com

**Data Availability Statement:** The data and R scripts used for statistical analysis are available at https://doi.org/10.5281/zenodo.11636401.

## Abstract

Extensive studies have shown that humans process faces holistically, considering not only individual features but also the relationships among them. Knowing where humans and dogs fixate first and the longest when they view faces is highly informative, because the locations can be used to evaluate whether they use a holistic face processing strategy or not. However, the conclusions reported by previous eye-tracking studies appear inconclusive. To address this, we conducted an experiment with humans and dogs, employing experimental settings and analysis methods that can enable direct cross-species comparisons. Our findings reveal that humans, unlike dogs, preferentially fixated on the central region, surrounded by the inner facial features, for both human and dog faces. This pattern was consistent for initial and sustained fixations over seven seconds, indicating a clear tendency towards holistic processing. Although dogs did not show an initial preference for what to look at, their later fixations may suggest holistic processing when viewing faces of their own species. We discuss various potential factors influencing species differences in our results, as well as differences compared to the results of previous studies.

## Introduction

Domestic dogs seem to pay a lot of attention to human faces, a behavior that can likely be attributed to their longstanding and close inter-species relationship with humans [1]. Consequently, there has been a substantial amount of research examining how dogs process human faces, revealing that their abilities appear to be similar to those of humans. Studies have reported that dogs can distinguish between unfamiliar and familiar human or dog faces [2–4], as well as discern emotional expressions on human faces [5, 6]. In their attempts to decipher facial information such as identity and emotion, dogs appear to utilize a holistic processing strategy similar to that of humans [7].

Holistic face processing, often considered a subset of configural face processing [8], has undergone extensive investigation in humans [9–15] and, to a lesser extent, in non-human animals, spanning reptiles, fish, and insects [16–19]. This method involves treating central

**Funding:** The study was funded by the Vienna Science and Technology Fund (WWTF, www.wwtf.at) No. CS11-026 awarded to ZV and in part by No. CS11-005 awarded to LH. The funders had no role in study design, data collection and analysis, decision to publish, or preparation of the manuscript.

**Competing interests:** The authors have declared that no competing interests exist.

inner facial features, such as the eyes, nose, and mouth, as a unified unit. In this configuration, the eyes and the mouth serve as boundaries, forming a configuration that largely resembles an inverted trapezoid. Studies on humans consistently reveal striking behavioral manifestations of holistic face processing. For example, when individuals view a face stimulus with its standard configuration of the inner facial features disrupted—such as when the typical inverted trapezoid configuration is rotated 180˚—their ability to recognize facial information deteriorates [9, 20]. Moreover, there is also a well-established understanding that humans possess specialized brain regions for processing faces, for example the fusiform face area [21], but see [22]]. Notably, a recent study has shown that these higher-level visual areas, as opposed to earlier visual regions like the primary visual cortex, display distinct responses to upright versus inverted face stimuli, indicating their central role in holistic face processing and subsequent recognition of facial information [23].

Studies with Japanese macaques or chimpanzees have shown that not only humans, but also the non-human primates use holistic processing strategy when detecting faces, indicating such strategy might be shared in non-human animals [24, 25]. However, the clarity and consistency of behavioral manifestations of holistic processing in non-primate animals, including dogs, remains uncertain (see [18] for a comprehensive discussion). Furthermore, while several studies have reported the presence of face-selective regions in dogs [26–28], a more recent study using more strictly controlled stimuli contradicted these earlier findings, emphasizing the need for further research on this subject [29].

## First fixation location of humans on faces and its implication for holistic processing

Another consistent behavioral manifestation of holistic processing in humans is the location of the first fixation on faces. On average, when detecting faces in peripheral vision, human individuals predominantly direct their first fixations toward the central face area, i.e., the inverted trapezoid configuration bounded by the eyes and mouth [30–44]. This fixation behavior appears to be universal, seemingly automatic (hard to inhibit), showing minimal influence from subjects' cultural backgrounds and the nature of recognition tasks, and is consistently observed across various real-life settings [37–40]. The above investigations have however offered distinct descriptions of the first fixation area, ranging from "on the nose" [31, 42, 44], "around the center of the nose" [30, 36], "a midline area just below the eyes" [33–35, 38, 41], "a featureless region between the eyes and the nose' [37], "the infraorbital margin" [32], "the area of the face midline (between the eyebrows, nose, and mouth)' [43], "the center of the image (including the nose, the rest of the face, and between the eyes)' [39] to "between the eyes and nose tip" [40].

It is well-known that the eyes and the mouth are crucial features for identifying facial information [45–50]. Then, what may drive this tendency to focus first on this predominately nasal region? It has been suggested that looking at this region is not necessarily intended to prioritize the detailed view of the nose ridge, but rather to efficiently capture a holistic representation of the crucial inner facial features with a single fixation [36]. In other words, by locating the gaze on the nose, one can quickly achieve relatively good, though not maximum, visual acuity for the view of both eyes and the nasion while obtaining comparable acuity for other crucial facial elements, such as the forehead, eyebrows, and the mouth located farther away. As a consequence, this fixation location has been labeled as an optimal viewing position or optimal first fixation position for humans which enables efficient and comprehensive processing of inner facial features in a single glance [30, 34]. This proposition is further supported by computer simulation models, such as the foveated ideal observer by [33]. Their model successfully

emulates human viewing behavior to faces by simultaneously integrating information from all inner facial features while considering the decline in resolution and sensitivity with increasing eccentricity [33, 35, 38].

## Evaluating discrepancies: Factors influencing the results of where humans look first and the longest on faces

The findings reviewed above regarding the first fixation position of humans contrast with the earlier consensus that the eyes are the initial focus of humans when viewing faces [51–54]. What factors contribute to this discrepancy? Notably, both [30, 33] attribute their distinct findings to variations in the face presentation location. Diverging from the studies aligning with the prior consensus, the investigations of Hsiao and Cottrell (2008) [30] and Peterson and Eckstein (2012) [33] presented a face positioned away from the starting gaze position. Indeed, the position of stimuli in relation to the starting gaze position in eye-tracking experiments is an acknowledged parameter that can influence the eye movements on the stimuli, potentially affecting the analysis and conclusions of the study [55]. For instance, when subjects have their initial gaze fixed at the center of the screen and a face stimulus is presented at the same location, the overlapping locations may allow them to perceive crucial facial details even before making their first eye movement. Subsequently, their initial fixation is likely to land on areas other than the central face region of the inverted trapezoid, as refixating that location would be redundant [32].

Another factor that may significantly contribute to the difference in results is how each study delineates the region of the eyes on face stimuli. A commonality among the studies that find that humans first look at the eyes is the generous space allotted to the facial area designated for the eyes. In these studies, the Area of Interest (AOI) for the eyes not only includes the eyes themselves but also significant surrounding areas, such as the upper nose and the infraorbital margin. This means that the upper nose area included in such spacious eyes AOI in those studies belongs to an "eyes" or "eye region" AOI, instead of a "nose' or "nose region' AOI, despite it actually being part of the nose. As a result, these studies are not able to exclude the possibility that what they classified as looks to the eyes were in fact fixations on the nose region. In contrast, the studies reporting the first fixations to be on the inverted trapezoid typically used either a more restricted AOI for the eyes or analysis methods, such as fixation maps that do not require AOI designation. The findings and insights from these studies emphasize the significance of how facial areas are delineated when interpreting fixation results and consequent conclusions in studies. Moreover, they prompt questions about whether the fixations in the studies that report that humans first look at the eyes or the eye region would still be classified as fixations on the eyes if less expansive AOIs for the eyes or alternative analysis methods were used.

Investigations of which facial region is fixated the longest would be equally influenced by the same issue. In studies where expansive AOIs for the eyes were employed, along with coinciding starting gaze and face stimulus positions, findings suggest that humans tend to fixate longest on the eyes [51–54]. Unfortunately, the majority of studies that found that the first fixation position was on the inverted trapezoid area typically presented results for only the initial few fixations [30–32, 34, 36, 39]. Therefore, it remains uncertain whether further fixations would be maintained on the inverted trapezoid area or gaze would shift elsewhere later in the trial. The study by Hessels et al. (2016) [56], which examined the impact of AOI production methods on eye movement results, offers valuable insights into this issue. In their investigation, where participants viewed a face for approximately four seconds, they found that AOI design methods excluding much of the upper nose area from the eyes AOI (hand-drawing and

grid methods) resulted in the longest dwell time for the nose AOI. In contrast, with other AOI methods, the longest dwell time was for the eyes AOI.

## Dogs' gaze behavior when viewing faces: Similar challenges in determining where dogs fixate first and the longest

Considering the findings in human studies, it seems useful to examine the locations of the first and subsequent fixations separately in dogs as well. Where do domestic dogs fixate first and the longest when observing human faces? Given that dogs possess an area centralis similar to the human fovea [57, 58] along with their successful history of social interaction with humans and well-documented facial information processing skills, it is reasonable to hypothesize that domestic dogs might exhibit gaze behavior akin to that of humans. They may fixate first and the longest on either the optimal first fixation position or on the eyes themselves. Several eye-tracking studies have explored dogs' eye movement behavior during the viewing of human faces [59–64], see [65] for a review focused on methodology. However, only a few of these studies have investigated the location of the first fixation on faces, and also their findings are conflicting. While [60] reported that the eyes (or the area around the eyes) were the most probable first fixation targets regardless of facial expression, [62] found that pet dogs most often (33.8%) fixated first on the area of the rest of the face, excluding the forehead, the eyes, and the mouth. Regarding further fixations, some dog studies indicated that dogs fixated on the eyes for the longest duration, but this observation was contingent on various experimental conditions, including comparisons such as inverted vs. upright faces, familiar vs. unfamiliar faces, oxytocin vs. placebo-treated dog subjects, and pet vs. lab or kennel dogs [59–64]. Although interesting, these factors complicates providing a simple answer to where dogs look first and longest when observing a (neutral) human or dog face.

Moreover, previous dog eye-tracking studies seem to face similar methodological issues as human eye-tracking studies. The majority of these dog studies adopted a very broad designation for the eyes AOI, so that the area for this AOI encompassed not only the individual eyes themselves but also a substantial portion of the surrounding regions (illustrative examples are shown in Fig 1 and detailed further in Table 1). Additionally, as shown in Table 1, most of these studies employed a procedure where the starting gaze position and the face stimuli were both at the center of the screen. Furthermore, some of the previous dog studies employed very short presentation durations, such as 1.5 s. Following the results of our previous investigation revealing that the duration of a dog's fixation is approximately four times longer than that of humans [66], 1.5 s is probably too short to produce results that can be comparable to those of

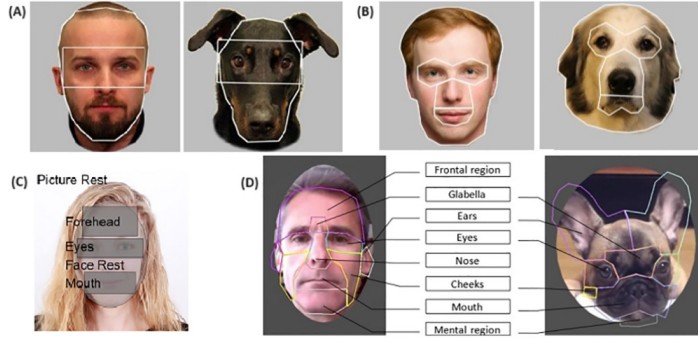

**Fig 1. Examples of AOI designs used in four eye-tracking studies with dogs.** Note that all four designs include either the infraorbital margin or upper nose area in their eyes AOI. Images are from A: [59], B: [60], C: [62], D: [64].

**Table 1. The eyes AOI design and locations of the starting position and stimulus presentation of six dog eye-tracking studies that investigated where dogs look on faces.**

| Study | AOI design for the eyes | Starting position | Face stimulus location | Presentation duration |
|---|---|---|---|---|
| [59] | expansive AOI for the eyes (AOI Eye area) in Fig 1 | screen center | screen center | 1.5 s |
| [60] | expansive AOI for the eyes (eyes AOI) in Fig 1 | screen center | screen center | 1.5 s |
| [61] | unclear, no example of eye region provided | screen center | left or right side of the screen | 7 s |
| [63] | no image no detailed description | screen center | screen center | 4 s |
| [62] | large AOI for the eyes (eyes AOI) in Fig 4 | screen center | screen center | 5 s |
| [64] | large AOI for the eyes (AOI Eye region) | screen center | screen center | 5 s to 12 s |

humans, as it would barely allow for one full fixation for dogs. These factors further obscure a clear understanding of where dogs first and predominantly fixate on a face as well as the extent to which they use a holistic processing strategy in comparison to humans.

## Our experiment

The above discussions highlight that interpreting the results regarding fixation locations on faces are sensitive to various factors, including AOI design, analysis method, starting gaze position, and stimulus presentation duration. This underscores the need for a new dog eye-tracking study that considers these factors. Therefore, we conducted an experiment to identify the specific areas on faces where humans and dogs fixate first and the longest, with the following experimental details. Firstly, in our study, we deliberately positioned our stimuli away from the central starting point for each trial. This ensured that only after the first saccade did the stimulus come into foveal view of the subjects. Secondly, we presented the face stimuli for seven seconds, which allowed both humans and dogs to make several fixations. Thirdly, we defined the eyes AOI to tightly encompass the eyes, while introducing a separate AOI tailored to the central facial area consisting of the inverted trapezoid positioned below the eyes. Finally, we included both dogs and humans as subjects and utilized an identical experimental setup, allowing for direct comparison of the results— a feature absent in previous canine eye-tracking studies of face processing (but see [64]). While not extensively studied, the predisposition of humans to initially fixate on the central area of the inner facial features may potentially extend to the faces of non-human entities, especially in cases where the face belongs to an animal species having significant social interaction with humans, such as domestic dogs. Moreover, research suggests that this predisposition might also apply to non-face objects, depending on stimulus attributes and the nature of visual tasks [67–70]. To explore these aspects in both species, we utilized a stimulus set comprising human faces and dog faces, along with non-face objects.

Drawing on the previous work reviewed above, we predicted that both humans and dogs, if they employ a holistic processing strategy, would most likely fixate first on the central inverted trapezoid area across all three types of stimuli. Furthermore, considering the varied findings from previous studies involving humans and dogs, we had an expectation that both species would predominantly fixate on either the central inverted trapezoid area or the eyes of human and dog faces for extended durations.

## Materials and methods

The eye movement data analyzed in this current paper constitute a subset of a larger dataset employed in previous studies by Park et al. (2020) [66] and Park et al. (2022) [65]. For a comprehensive understanding of the apparatus, experimental procedure, and dog training, readers

are referred to [66]. For a more detailed description of the experimental procedure and the application of the Nyström & Holmqvist (2010) algorithm [71] for fixation classification, refer to [65], which provides in-depth methodological insights into dog eye tracking.

## Ethics statement

All experimental protocols were approved by the institutional ethics and animal welfare committees of the University of Veterinary Medicine, Vienna and Medical University of Vienna in accordance with the GSP guidelines and national legislation (02/03/97/2013 and 1336/2013, respectively). The participant recruitment period for this study was from May 15, 2013, to May 15, 2015. All methods were carried out in accordance with the relevant regulations and guidelines. A written informed consent was obtained from all adult human participants and dog owners before the experiment was conducted.

## Subjects

We recruited dog subjects by contacting dog owners who had previously agreed to participate in behavioral and cognitive studies in the lab. Initially, 33 dogs were recruited for pre-experiment training. Of these, eight dogs had vision-related morphological characteristics that interfered with the tracking ability of the eye-tracking system (for more details see [65]). These dogs were excluded from further data collection. A further ten dogs could not complete the pre-experiment training, thus 15 dogs were included in the data collection. 14 dogs (age: m = 5.57 years, SD = 2.88 years; sex: six males and eight females) could complete all trials. The dog subjects were one Akita Inu, four Border Collies, one Boxer, one Petit Brabancon, one Golden Retriever, two Siberian Huskies, one Jack Russell Terrier, one Parson Russell Terrier, one Rhodesian Ridgeback, and one mixed breed. However, not all dogs provided valid data for all dependent variables, thus the number of valid data differs per dependent variable (see Table 2 for the details).

In addition, 15 human participants (age: M = 29.2 years, SD = 10.5 years; gender: six males and nine females), volunteering graduate students, dog owners or university staff with normal or near normal vision without glasses (glasses were off during the experiment), completed all trials. The human subjects did not present with droopy eyelids or other morphological characteristics that might have interfered with the eye-tracker's performance.

## Apparatus

Dogs and humans both took part in experiments at the lab, using identical setups and an Eye-Link 1000 eye tracker (SR Research) equipped with a 35-mm camera lens mounted in desktop

**Table 2. Number and % of valid trials out of total trials per stimulus type (60 in humans and 56 in dogs) included in the statistical models.**

| Dependent variable | Stimulus category | Humans | Dogs |
|---|---|---|---|
| First fixation AOI hit probability | Human face | 43/60 (72%) | 14/56 (23%) |
| | Dog face | 43/60 (70%) | 20/56 (32%) |
| | Non-face object | 50/60 (83%) | 14/56 (22%) |
| First fixation-centroid distance | Human face | 43 (72%) | Not Applicable |
| | Dog face | 42 (70%) | Not Applicable |
| | Non-face object | Not Applicable | Not Applicable |
| Relative total fixation duration | Human face | 60 (100%) | 48 (80%) |
| | Dog face | 60 (100%) | 53 (88%) |
| | Non-face object | 60 (100%) | 48 (80%) |

mode. The gaze position of the right eye was captured at a rate of 1000 Hz from a distance of 50–55 cm. Pupil detection was conducted using centroid mode. Prior to recording, each dog underwent training to keep their head still on the chinrest and maintain direct gaze at the screen, as well as to fixate on calibration points (for training specifics, refer to [66]). Human participants were provided instructions to avoid head movements but did not undergo formal training. At the onset of each experiment, we calibrated the right eye of each participant until subsequent validation indicated an average error of less than 1.5˚. While most human subjects did not require recalibration, for the majority of dog subjects, we found it necessary to repeat the calibration and validation procedures at least once. Each trial commenced with a display featuring a starting fixation point at the center of the screen. Once the experimenter confirmed the participant's fixation on the point, a stimulus was presented for seven seconds, after which the second trial of a block began. Following every block of two trials, both dog and human participants were given the opportunity to move or receive a food reward. Therefore, we conducted recalibration (but without validation due to the limited attention span of dogs) before each block and also prior to the second trial of a block if there was noticeable movement, such as head rotation or deviation from the chinrest after the first trial. The average accuracy achieved during both training and experimentation was 1.56 degree for dogs and 1.09 degree for humans. Eye movement recordings were saved for offline analysis. Fixations were classified using a custom version of the Nyström & Holmqvist (2010) algorithm [71] implemented by Niehorster et al. (2015) [72] (available at https://github.com/dcnieho/NystromHolmqvist2010).

## Stimuli and experimental design

In total, 12 photos of human faces, dog faces and non-face objects (Fig 2) were shown to human and dog subjects onto a dark grey back-projection screen. The four human face photos with neutral facial expression were collected from the Radboud FACE Database [73]. The four dog face photos were collected from internet websites with their permission. The four control, non-face object images were created by the main author in order to mimic the overall color and composition of the face stimuli (bigger element or more elements on the upper half of the objects). The sizes of images were kept similar within each stimulus category. The overall size of human face photos was 246 (w) x 341 (h) pixels, 411 (w) x 306 (h) pixels for dog faces, and 257 (w) x 257 (h) pixels for non-face objects which corresponds to similar viewing angles of approximately 8˚ (w) x 10˚ (h) for human faces, 12˚ (w) x 9˚ (h) for dog faces, and 8˚ (w) x 8˚ (h) for non-face objects. This makes the viewing angle of the images comparable to that of a real human face at one meter, the distance of usual social interaction [30]. Importantly, we carefully adjusted image configurations and presentations to address two known biases in

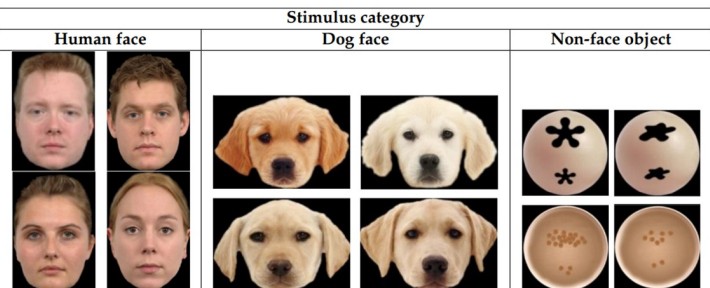

**Fig 2. The twelve stimuli presented to each dog and human subject in this study.**

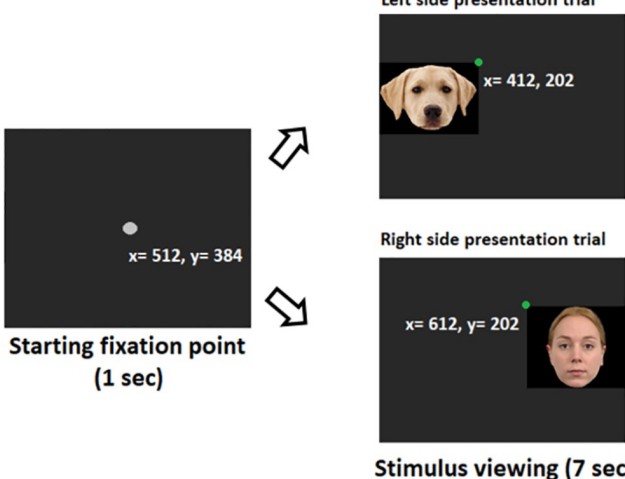

**Fig 3. Stimulus presentation scheme of two different example trials.** In each trial, a stimulus was displayed either to the left or right of the central starting fixation point on the screen (x = 512, y = 384). The x-coordinate of the upper vertex of the nearer edge (indicated by a green dot) of the (invisible) image frame was positioned 100 pixels away from the starting fixation point. Each stimulus was presented for seven seconds.

human stimulus viewing: scene-center and screen-center biases [73]. To counteract scene-center bias, we used a black background for the images, ensuring the background seamlessly blended with the dark grey screen, so that only the object, not the rectangular frame of an image, was visible to subjects. To tackle screen-center bias, each stimulus was presented 100 pixels off center on either the left or right side of the screen (see Fig 3). Each participant viewed six blocks, each containing two trials. Unique random sequences of the 12 photos were created for each subject. Presentation of the starting point and stimuli was controlled by an executable file generated by the Experiment Builder (version 1.10.165) of SR Research.

## Statistical analysis

For all statistical tests, given our repeated measures with between-subject factor design, we used (Generalized or General) Linear Mixed-Effects Models [75]. Each model was devised to test data of both species in each stimulus condition. The specifications of the models are summarized in Table 3. An $\alpha$-level of 0.05 was used to judge the significance of the model parameters (type III ANOVA using Wald chi-square tests) during model comparisons, the significance of terms included in the final models, or the significance of differences during

**Table 3. Summary of the statistical models tested.**

| Dependent variable | Distribution Type | Link function | Factor specification | Model fitness statistics | | |
|---|---|---|---|---|---|---|
| | | | | Stimulus category | Marginal $R^2$ | Conditional $R^2$ |
| First fixation AOI hit probability (0 < probability < 1) | Binomial | logit | species + AOIs + species:AOIs + (1 \| name) | Human face | 0.98 | 0.98 |
| | | | | Dog face | 0.97 | 0.97 |
| | | | | Non-face object | 0.97 | 0.97 |
| Relative total fixation duration on AOIs (0 < rate < 1) | Binomial | logit | species + AOIs + species:AOIs + (1 \| name) | Human face | 0.89 | 0.89 |
| | | | | Dog face | 0.88 | 0.88 |
| | | | | Non-face object | 0.28 | 0.28 |

**Table 4. Summary of the model test results (type III sums of squares ANOVA using chi-square tests).**

| Stimulus category | Eye movement responses | Tested factors in the model | | |
|---|---|---|---|---|
| | | Species | AOIs | Species x AOIs |
| Human face | First fixation AOI hit probability | $X^2(1) = 6.67$ $p = .0098$ $(S = 6.68)$ ** | $X^2(4) = 5.54$ $p = .24$ | $X^2(4) = 18.91$ $p = .0008$ $(S = 10.26)$ *** |
| | Relative total fixation duration on AOIs | $X^2(1) = 13.31$ $p = .0003$ $(S = 11.89)$ *** | $X^2(4) = 42.27$ $p < .0001$ $(S = 26.02)$ **** | $X^2(4) = 13.37$ $p = .0096$ $(S = 6.70)$ ** |
| Dog face | First fixation AOI hit probability | $X^2(1) = 13.78$ $p = .0002$ $(S = 12.25)$ *** | $X^2(4) = 4.13$ $p = .39$ | $X^2(4) = 10.69$ $p = .03$ $(S = 5.04)$ * |
| | Relative total fixation duration on AOIs | $X^2(1) = 0.46$ $p = .5$ | $X^2(4) = 29.83$ $p < .0001$ $(S = 17.53)$ **** | $X^2(4) = 2.29$ $p = .68$ |
| Non-face control | First fixation AOI hit probability | $X^2(1) = 1.75$ $p = .19$ | $X^2(4) = 3.52$ $p = .47$ | $X^2(4) = 5.94$ $p = .20$ |
| | Relative total fixation duration on AOIs | $X^2(1) = 0.03$ $p = .86$ | $X^2(4) = 11.86$ $p = .02$ $(S = 5.76)$ * | $X^2(4) = 5.80$ $p = .22$ |

chance level comparisons and pairwise comparisons (one-tailed $Z$) [76] (significance code: 0 < '****' < 0.0001 < '***' < 0.001 < '**' < 0.01 < '*' < 0.05). As suggested by [77], $p$-values less than 0.05 in Table 4 are accompanied by an associated surprisal value (Shannon information), denoted as $S = -\log_2(p)$ (The $p$-values used in this calculation were taken before rounding). This value represents the amount of surprise linked to the given $p$-value, as compared to coin toss results [77]. For instance, a $p$-value of 0.05 is akin to obtaining 4 heads or tails in 4 consecutive coin tosses. Therefore, higher surprisal values indicate greater evidence against the hypothesis being tested. Where appropriate, effective size $d$, calculated using `eff_size` function (emmeans), is reported. All tests were conducted in R (R Core Team, 2019) version 4.0.0 using the lme4, anova, car, effect, emmeans, MuMIn, glmmTMB, stats, and other required packages. The data and R scripts used for statistical analysis are available at https://doi.org/10.5281/zenodo.11636401 [78].

**AOIs.** We defined five distinct Areas of Interest (AOIs) for each stimulus, as outlined in Fig 4. These AOIs were labeled as "eyes", "around eyes", "inner features center", "else upper half", and "else lower half". The eyes AOI precisely covered the region outlining the visible part of both eyes and a portion of the nasion in-between the two eyes. The inner features center AOI took on the shape of an inverted trapezoid, extending from just below the eyes AOI to the upper lip's edge. The around eyes AOI referred to the region surrounding the eyes AOI, excluding the overlapping portion with the eyes AOI and the inner features center AOI. The else upper half AOI and the else lower half AOI encompassed the remaining upper or lower half of the face or object, not covered by other AOIs. For non-face object stimuli, the eyes AOI and the around eyes AOI were referred to as the upper features AOI and the around features AOI, respectively. Note that AOI size was not included as a covariate in our statistical models, as doing so was not supported by model selection procedures and did not change the overall conclusions of our study.

**Exclusion of invalid data.** A total of 180 trials were recorded for the human participants and 168 trials for the dogs. For both dependent variables, the first fixation AOI hit probability and the relative total fixation duration, we excluded the data of trials where there was no fixation on the stimulus: 19 trials in dogs and none in humans. Further, despite monitoring by the experimenter, in dogs as well as in humans, it happened that the participants' eyes were not on the central starting point before the stimulus. Therefore, for the first fixation AOI hit probability data, we removed the data of trials in which the starting gaze position of the subject was

**Fig 4. Examples of AOIs and whole stimulus area for each stimulus condition.** Each AOI is indicated as a red shape overlaying a part of each stimulus. The size of each AOI is indicated in (horizontal) visual angle below each AOI and the centroid of each of the 2 areas (the inner features center AOI and whole stimulus) is indicated as a solid yellow point. The relative size of the stimuli are not actual.

more than 4˚ of visual angle (133 pixels) away from the point (see Table 2 for the number of trials included in the statistical analyses).

**Analysis of first fixation AOI hit probability: Which AOI did dogs and humans most likely fixate on first?.** In each trial, we computed the probability of the first fixation landing on each AOI. For the data of each stimulus condition, a Generalized Linear Mixed-Effects model was used including the following terms: species, AOIs, and their interaction (Table 3). With five AOIs in consideration, the chance level probability—where any one AOI among the five AOIs is hit at random—was 0.2. To assess if the estimated first fixation AOI hit probability of each AOI is significantly larger than chance, the estimate of each AOI was compared to this chance level using one-sided t-tests. If the probability of the first fixation landing on a given AOI was higher than chance, it was further compared to the probabilities of the other four AOIs (pairwise comparisons) to see if the differences between all AOI pairs are significant. $p$ values of the chance level and pairwise comparisons were adjusted using the Bonferroni correction method.

**Analysis of first fixation-AOI centroid distance: Were human first fixations closer to the centroid of the inner features center AOI than to that of the whole stimulus?.** This analysis was conducted based on the results of the first fixation AOI hit probability. As we report below, it was observed that humans most likely first fixated on the inner features center AOI of human and dog faces, but not of non-face objects, whereas dogs did not exhibit a clear AOI preference for their first fixations in any of the stimulus conditions. Thus only the first fixation data of humans in the human face and dog face conditions are included in this analysis. We assessed if those first fixations of humans on the inner features center AOI of human and dog faces indeed targeted the center of the inner features center AOI, instead of mere center of the faces, e.g., due to a "scene-center bias" [74]. We assumed that a fixation that targeted the inner features center AOI would be located closer to the centroid of the AOI than the centroid of the whole face stimulus. To establish the centroids (x and y pixel coordinates) of the two

areas, we used the regionprops function of Matlab. The pixel coordinates of the first fixations were extracted from the fixation data classified by the Nyström & Holmqvist (2010) algorithm. The pixel distances from the first fixation to the two centroids were computed using the formula:

$$d = \sqrt{(x_2 - x_1)^2 + (y_2 - y_1)^2}$$

Within each face condition, the two distances were compared against each other by means of a one-sided t-test.

**Analysis of relative total fixation duration on AOIs: Which AOI did dogs and humans fixate the longest?.** The relative total fixation duration for each AOI was computed by dividing the total duration of fixations on a specific AOI by the total duration of fixations on all five AOIs in a trial. A Generalized Linear Mixed-Effects model was used for each stimulus condition which includes species, AOI and their interaction (Table 3). With a total of 5 AOIs, the chance level relative total fixation duration—representing the relative duration when all five AOIs are fixated equally long—was 0.2. To assess if the estimated relative total fixation duration of each AOI is significantly larger than chance, each estimate associated with an AOI was compared to this chance level using one-sided t-tests. Further, if the estimate of an AOI was higher than chance level, it was further compared to the estimates of the other four AOIs (pairwise comparisons) to see if the differences between all AOI pairs are significant. $p$ values of the chance level and pairwise comparisons were adjusted using Bonferroni correction method.

## Results

### First fixation AOI hit probability: Humans, but not dogs preferentially first fixated on the inner features center AOI for both human and dog faces

In the human face condition (Fig 5A, Table 4), the probability that the first fixation landed on a specific AOI was overall significantly different between species, and there was an interaction between species and AOIs. When observing human faces, humans predominantly fixated first on the inner features center AOI significantly more than expected by chance ($Z = 6.69$, $p < .0001$), and more than on the other four AOIs (eyes: $Z = 5.95$, $p < .0001$; around eyes: $Z = 8.45$, $p < .0001$; else upper half: $Z = 9.15$, $p < .0001$; else lower half: $Z = 7.84$, $p < .0001$). On the contrary, dogs predominantly fixated first on the else upper half AOI more than expected by chance ($Z = 3.10$, $p = 0.005$). However, the probability of first fixating the else upper half AOI was not significantly different from that of the inner features center AOI ($Z = 1.6$, $p = 0.44$), which complicates seeing a clear preference similar to that in humans.

In the dog face condition (Fig 5B, Table 4), the probability that the first fixation landed on a specific AOI was overall significantly different between species, and there was an interaction between species and AOIs. When observing dog faces, humans predominantly fixated first on the inner features center AOI significantly more than expected by chance ($Z = 7.30$, $p < .0001$), as well as more than the other four AOIs (eyes: $Z = 11.37$, $p < .0001$; around eyes: $Z = 13.72$, $p < .0001$; else upper half: $Z = 13.72$, $p < .0001$; else lower half: $Z = 8.49$, $p < .0001$). Dogs did not exhibit a clear preference of their first fixation for any AOI over chance level (inner features center: $Z = 1.11$, $p = 0.67$; eyes: $Z = -0.56$, $p = 1.00$; around eyes: $Z = 1.11$, $p = 0.67$; else upper half: $Z = -0.56$, $p = 1.00$; else lower half: $Z = -1.1$, $p = 1.00$).

In the control condition (Fig 5C, Table 4), the first fixation AOI hit probability was not significantly affected by any of the terms. When presented with non-face objects, humans predominantly fixated first on the inner features center AOI significantly more than expected by chance ($Z = 5.71$, $p < .0001$), yet the probability for this AOI was not significantly different

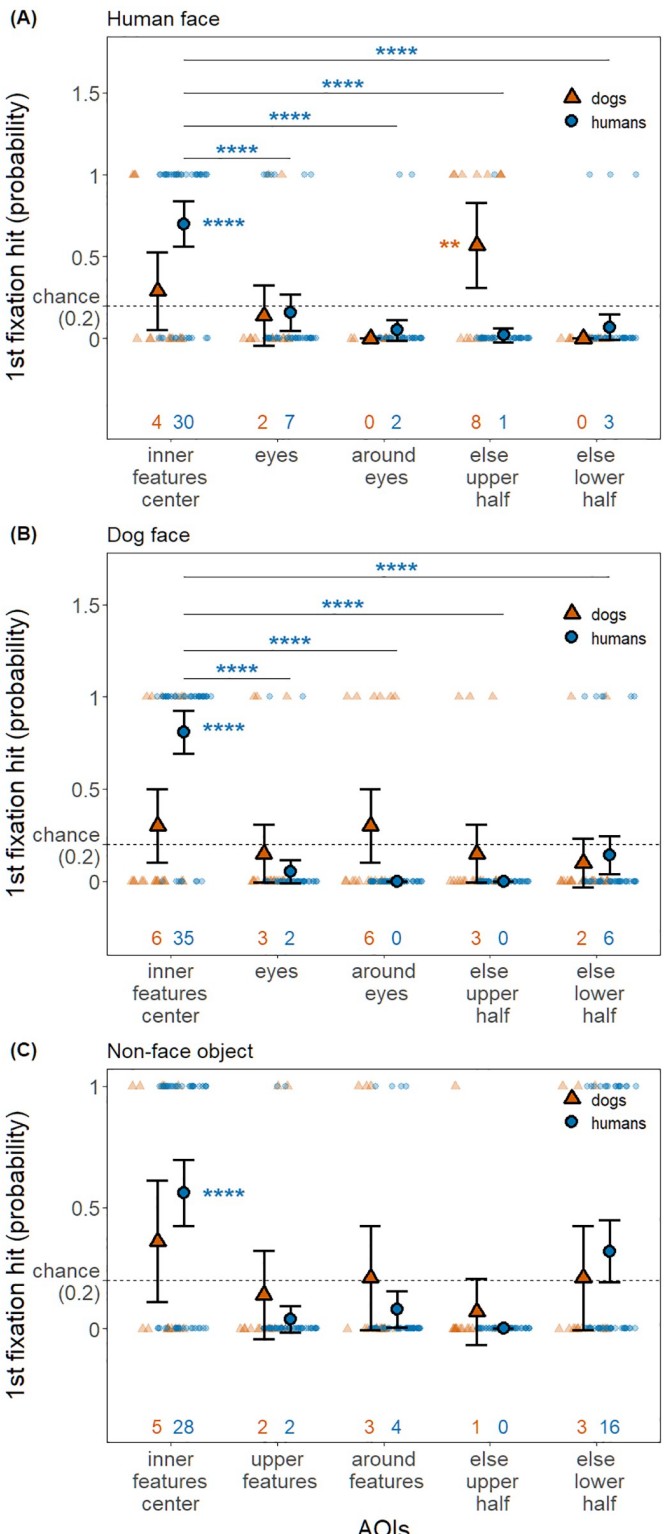

**Fig 5. The probability that the first fixation of humans and dogs was on one of the five AOIs in the three stimulus conditions.** Each panel corresponds to each stimulus condition (panel A: human face; panel B: dog face; panel C: non-face object). Solid symbols (a triangle or a circle) and error bars depict estimated means and 95% confidence intervals, respectively. Smaller symbols of the same shape in softened color in the background indicate trial data used for the analysis. Numbers on the x-axis indicate the number of hits on the corresponding AOI across dog/human participants

and trials. Dashed line indicates chance level probability (0.2) and AOIs with the estimate significantly higher than the chance level is indicated with stars next to the solid symbols. Further, the AOI pairs with significant differences are annotated with lines and stars, yet only if the differences between all AOI pairs are significant.

from that of the else lower half AOI ($Z$ = 2.49, $p$ = 0.051), making it challenging to confirm a clear preference. Dogs, on the other hand, did not exhibit a clear preference of their first fixation for any AOI over chance level (inner features center: $Z$ = 1.43, $p$ = 0.38; upper features: $Z$ = -0.53, $p$ = 1.00; around features: $Z$ = 0.13, $p$ = 1.00; else upper half: $Z$ = -1.14, $p$ = 1.00; else lower half: $Z$ = 0.13, $p$ = 1.00).

Further, we examined the results of the first fixation-centroid distance measure to assess whether the first fixations of humans on the inner features center AOI of human faces and dog faces indeed targeted the center of the AOI instead of the center of the whole face area driven by scene-center bias. The first fixations of humans were significantly closer to the centroid of the inner features center AOI than the centroid of the whole face area of both human and dog faces (human face: t(70.3) = 2.78, $p$ = 0.0035, $d$ = 0.6, 95% CI [0.15, 1.05]; dog face: t(68.3) = 2.67, $p$ = 0.0048, $d$ = 0.58, 95% CI [0.12, 1.05]) (Fig 6).

## Relative total fixation duration: Humans preferentially fixate the inner features center AOI of both human and dog faces, while dogs did so only on dog faces

In the human face condition (Fig 7A, Table 4), the relative total fixation duration was significantly different between species and among AOIs, and there was an interaction between species and AOIs. When observing human faces, humans predominantly fixated on the inner features center AOI significantly more than expected by chance ($Z$ = 4.29, $p$ < .0001) as well as more than on the other four AOIs (eyes: $Z$ = 3.33, $p$ = 0.004; around eyes: $Z$ = 6.31, $p$ < .0001; else upper half: $Z$ = 6.77, $p$ < .0001; else lower half: $Z$ = 5.12, $p$ < .0001). On the other hand, dogs fixated predominantly on the else upper half AOI, significantly more than chance ($Z$ = 5.63, $p$ < .0001) as well as more than on the other four AOIs (inner features center:

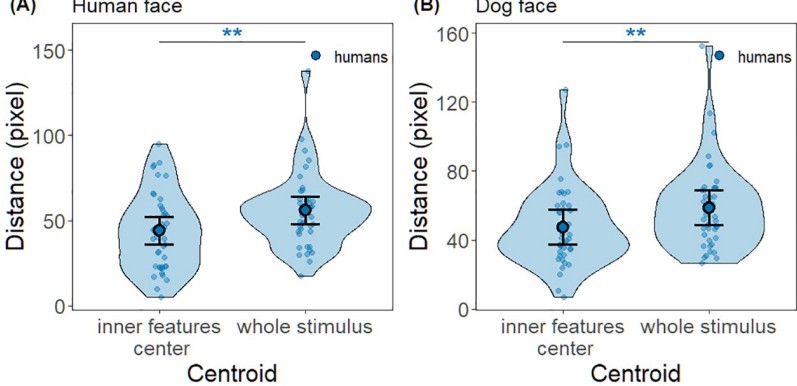

**Fig 6. Pixel distances between the first fixations of humans and the centroid of one of the face areas.** Each violin plot, depicting kernel probability density, shows the distribution of the measured pixel distances: the inner features center AOI or the whole stimulus area. Note that in both human face and dog face conditions, the first fixations of humans were on average significantly closer to the centroid of the inner features center AOI than that of the whole stimulus area. Solid circle and error bars overlaid on the violin plots depict estimated means and 95% confidence intervals, respectively. Dots indicate trial data included in the analysis.

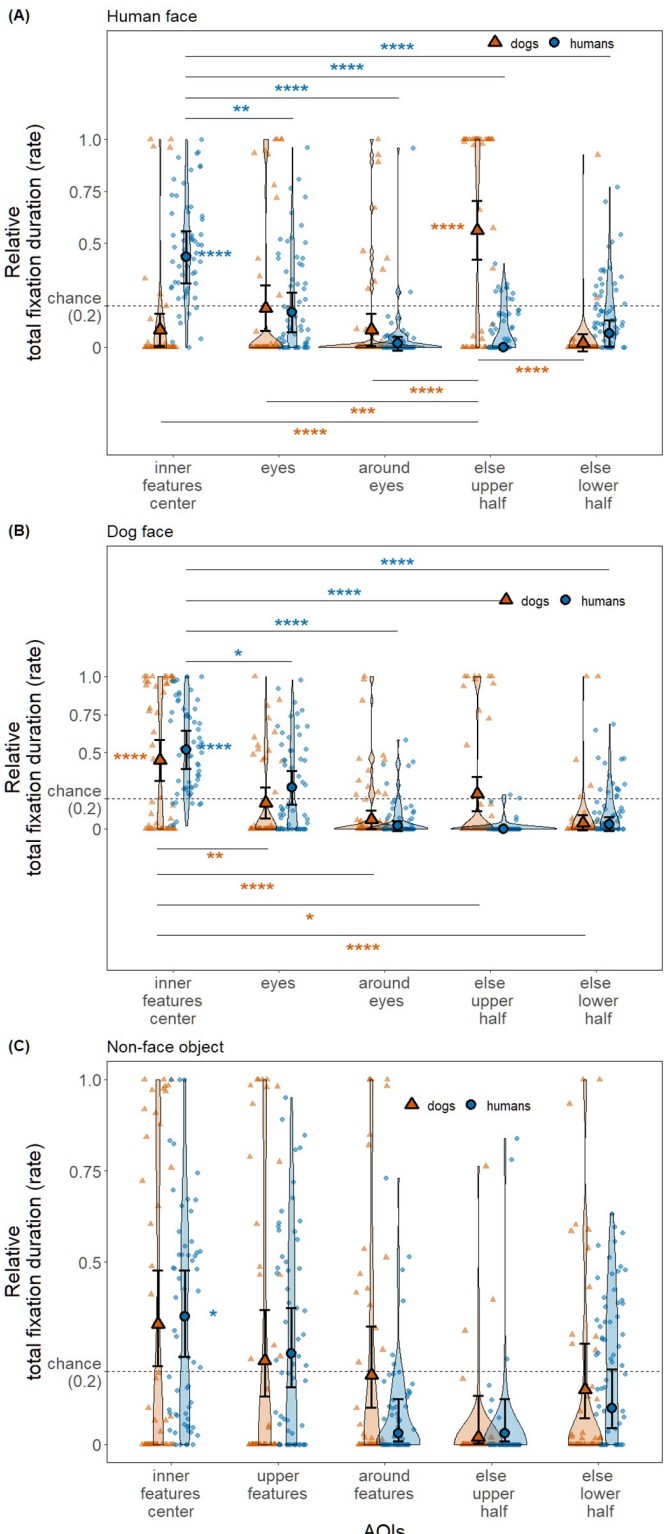

**Fig 7. The distribution pattern of relative total fixation duration (rate) of humans or dogs.** Each panel has a plot of each stimulus condition (panel A: human face; panel B: dog face; panel C: non-face object). Solid circle and error bars overlaid on the violin plots depict estimated means and 95% confidence intervals, respectively. Smaller symbols of the same shape in softened color in the background indicate trial data used for the analysis. Dashed line indicates chance level rate (0.2) and AOIs with the estimate significantly higher than the chance level are indicated with stars next to the

solid symbols. Further, the AOI pairs with significant differences are annotated with lines and stars, yet only if the differences between all pairs in the stimulus condition are significant.

$Z = 5.85$, $p < .0001$; eyes: $Z = 4.12$, $p = 0.0002$; around eyes: $Z = 5.85$, $p < .0001$; else lower half: $Z = 7.27$, $p < .0001$).

In the dog face condition (Fig 7B, Table 4), the relative total fixation duration was significantly different only between AOIs. When observing dog faces, humans predominantly fixated on the inner features center AOI significantly more than expected by chance ($Z = 5.62$, $p < .0001$) as well as more than on the other four AOIs (eyes: $Z = 2.90$, $p = 0.015$; around eyes: $Z = 7.51$, $p < .0001$; else upper half: $Z = 8.01$, $p < .0001$; else lower half: $Z = 7.05$, $p < .0001$). Similarly, dogs predominantly fixated on the inner features center AOI significantly more than expected by chance ($Z = 4.34$, $p < .0001$) as well as more than on the other four AOIs (eyes: $Z = 3.31$, $p = 0.004$; around eyes: $Z = 5.26$, $p < .0001$; else upper half: $Z = 2.54$, $p = 0.045$; else lower half: $Z = 5.67$, $p < .0001$). It is notable that in comparison to the results of human face condition, in this condition the differences in the relative total fixation duration between the most favored AOI (the inner features center AOI in both species) and the second most favored AOI (humans: eyes AOI, dogs: else upper half AOI) are less in both species.

In the non-face object condition (Fig 7C, Table 4), the relative total fixation duration was significantly different only between AOIs. Humans fixated preferentially only on the inner features center AOI, significantly more than chance ($Z = 2.84$, $p = 0.01$), yet not significantly more than upper features AOI ($Z = 1.20$, $p = 0.92$), making it challenging to confirm a clear preference. Dogs did not preferentially look more to any AOI than expected by chance (inner features center: $Z = 2.26$, $p = 0.06$; upper features: $Z = 0.51$, $p = 1.00$; around features: $Z = -0.22$, $p = 1.00$; else upper half: $Z = -2.44$, $p = 1.00$; else lower half: $Z = -0.93$, $p = 1.00$).

## Discussion

### Our predictions and limitations

In terms of the first fixation AOI hit probability measure, our expectation was that both humans and dogs would primarily fixate on the inner features center AOI, with these fixations being closest to the centroid of the AOI across all three stimulus conditions. While our predictions generally held true for humans, with deviations observed only in the non-face object condition, the results for dogs did not align with our expectations. Regarding the measure of the relative total fixation duration, our prediction was that both humans and dogs would predominantly fixate on either the inner features center AOI or the eyes AOI of faces. Among humans, a distinct preference was observed for fixating on the inner features center AOI for both human and dog faces, persisting even after their initial fixations. In contrast, dogs exhibited this preference partially and only when viewing dog faces. However, before delving into the discussion of the results, it's important to note that the average accuracy in our study may not be optimal, particularly for dogs, whose accuracy was lower than that of humans. With this in mind, further discussion on our findings follows below.

### Humans: Strong holistic processing tendency of humans extending beyond their first fixations and human faces

Humans predominantly directed their first fixations to the inner features center AOI of both human and dog faces. Moreover, by employing the first fixation centroid distance measure, we have demonstrated that their first fixations were on average closer to the centroid of the inner features center AOI than that of the whole face, indicating that the first fixation locations of

humans were more consistent with holistic face processing, than with a scene-center bias. Further, our results of the relative total fixation duration unveiled that humans not only fixated first, but also for the longest on the inner features center AOI—a pattern that was also evident when viewing dog faces. Our results in humans echoes the findings of the previous human studies on the optimal first fixation position when viewing human faces [30, 32–34, 40]. Further, continuing with the notion that the inner features center AOI is the optimal location for holistic face processing, this finding might suggest that humans uphold their holistic processing inclination when viewing faces, extending well beyond their initial fixations and to faces of other species, such as dogs. Further, comparing the details of the results between the two face conditions, it seems that this extended holistic processing tendency in humans is strongest when viewing faces of their own species.

Our results are particularly intriguing, especially considering that, unlike in most human studies, both human and dog subjects in our experiment viewed a face stimulus freely, without any specific facial information recognition task assigned. This suggests the possibility that humans might naturally and habitually exhibit fixation behavior optimized for facial information recognition whenever they encounter a face—a behavior that holds significant evolutionary importance [17, 34, 79]. Support for this can be found in the findings of [80], where informing subjects about the recognition task did not significantly influence either their eye movement results or their face recognition performance [80]. Similarly, in Peterson et al. (2016)'s study, subjects exhibited similar first fixation patterns on faces in both real-world contexts with free viewing and laboratory facial information recognition tasks [38]. While there have been other studies investigating eye movements of humans processing non-human faces [81–84], to the best of our knowledge, our study is the first to investigate both the location of the first fixation and the location fixated the longest by humans when viewing non-human faces. The question remains whether humans would further generalize this fixation strategy for the faces of other animal species, or if it is specific to the faces of *Canis familiaris*, the only large carnivore species that humans have been domesticating for over 23,000 years [85]. Future studies employing face stimuli of a wider range of animal species could provide valuable insights into this.

When we compare our results with human studies that had face viewing durations similar to our experiment, the results do not coincide. These studies collectively reported that humans fixated predominantly on the eyes for the longest duration [51–54]. We attribute this discrepancy mainly to the differences in the design of the eyes AOI. In the human eye-tracking studies mentioned above, the AOIs for the eyes included a large margin surrounding the eyes themselves, often encompassing the area just below the eyes (but see also Figure 1 of [80, 86]). Unfortunately, direct comparisons of our results with studies using restrictive designation of the eyes like ours are not possible due to differing analysis methods and much shorter viewing durations in those studies compared to our seven seconds.

While our results indicate that humans do not predominantly fixate first or longest on the eyes themselves when viewing faces, this should not be misinterpreted as humans disregarding or failing to extract information from the eyes entirely. The eyes undeniably convey vital information regarding gaze direction, which is essential for social interaction [45, 49, 50, 87–91]. Probably due to this reason, similar to the first fixation behavior on the inverted trapezoid area, fixating on the eyes is not entirely controllable. This was demonstrated by studies such as [44, 86, 92], where human subjects struggled to avoid fixating on the eyes during "don't-look the eyes" tasks. Indeed, in our results, the eyes AOI was the second most fixated upon AOI following the inner features center AOI in both face conditions. Research has shown that although humans may not initially prioritize fixating on the eyes, their presence and location within the typical configuration of facial features significantly influence where initial fixations

are placed [79]. Therefore, our findings suggest that after their initial fixation, individuals do shift their focus to the eyes but subsequently return to the optimal fixation position. This approach allows them to maintain a comprehensive view of the entire inner facial features while having a parafoveal view of the eyes, facilitating the detection of crucial eye-specific information, such as gaze direction.

However, our prediction for the non-face object condition did not hold true for humans: compared to the results of both face conditions, it was unclear which AOI humans fixated on first and for the longest duration. We anticipated that humans would likely fixate first on the inner features center AOI, given that our non-face object stimuli roughly mimic faces: circular objects with inner elements primarily situated in their upper halves, creating a top-heavy appearance. The unclear eye movement strategy in humans for non-face objects in our study suggests that the optimal fixation strategy while viewing faces likely depends on finer characteristics of faces, such as the typical face outline and the presence of two upper elements instead of one or several upper elements as in our control images. It was similarly shown in the study of [79].

## Dogs: Less optimal gaze behavior of dogs on human faces and possible contributing factors

Our results indicate that dogs, unlike humans, did not predominantly fixate first on the inner features center AOI of either human or dog faces. The comparison between AOI pairs did not reveal a clear preference. The lack of distinct first fixation results in dogs may stem from various factors, with the relative scarcity of first fixation data in dogs likely being the most significant contributor. On the other hand, clearer results for the relative total fixation duration in dogs (to be discussed later) could be obtained, likely due to the comparable amount of data available for this variable as in humans.

We have compared our findings with those of [60], the study whose analysis method was overall the most similar to ours, and explored potential reasons behind any discrepancies. Unlike our findings regarding first fixations, their research revealed that dogs consistently tended to fixate on the eyes of human faces as their first target, regardless of the facial expression depicted [60]. We suspect that several differences between the studies may have significantly contributed to the difference in results.

First, our analysis applies chance level comparison. In our study we compared the result of each AOI to chance level and deemed outcomes to be not notable if the first fixation hit probability of an AOI was not significantly larger than chance. When applying the chance level comparison similarly to the results of [60], we could make the same conclusion as ours for their study, as no AOIs in their results had a probability higher than 0.33, the chance level calculated considering the three AOIs in their study. To the best of our knowledge, no prior dog eye-tracking studies have compared their dog eye movement results to chance level. Some may argue that assessing eye movement results of dogs against chance level is overly stringent. However, this analytical approach has been widely adopted and shown to be essential across various psychological studies of non-human animals for identifying meaningful patterns and deviations in their behavioral responses [93–96]. Similarly, it has been frequently employed in eye-tracking studies involving non-adult subjects with the same rationale [97–100]. Therefore, we do not consider it overly stringent, especially given that in our study we applied the chance level comparison equally to the results of both dogs and humans.

Second, as similarly discussed with human studies earlier, there is a difference between the studies in how the eyes AOI are designated. While our AOI for the eyes tightly encompasses the two eyes and the nasion, the counterpart in [60] has a more generous space surrounding

the eyes, which includes, importantly, a significant part of the upper area around the eyes, the upper nose area, and the infraorbital margin (Fig 1, Table 1). This difference likely would have resulted in more fixations labeled as fixations on the eyes in their study compared to ours. Moreover, the disparity in the presentation location of face stimulus between the studies might have also contributed to the observed differences.

Contrary to the first fixation results, we observed distinct preferences for AOIs in the relative total fixation duration results of dogs. Interestingly, while humans exhibited a consistent fixation preference for the inner features center AOI of both human and dog faces, the AOI preference of dogs differed between the two face types. Dogs predominantly fixated on the else upper half AOI of human faces, while they fixated on the inner features center AOI the longest when viewing dog faces, mirroring the holistic fixation pattern observed in humans. Based on these results, it can be suggested that dogs may apply a holistic processing strategy more effectively when observing dog faces compared to human faces. What might be the reason for this?

Firstly, we cannot exclude the possibility that this differential AOI preferences of dogs between dog and human faces could simply stem from the differences in the characteristics of the stimuli. For instance, the eyes and the nose of dog faces in our stimuli were overall larger relative to those of human face stimuli. Additionally, those inner facial features of dog faces had greater contrast against the background color of the faces compared to those of human faces. The larger and more distinct inner facial features of dog faces might have made it easier for dogs to select and fixate on the inner features center AOI than when viewing human faces.

Alternatively, could the distinct area of interest (AOI) preference observed in dogs be due to dogs exhibiting a more efficient use of the holistic processing strategy when viewing faces of their own species? Findings in human and non-human primate studies suggest that this hypothesis is not unfounded. A perceptual advantage towards own-species faces has been observed in many studies of humans and non-human primates [25, 101, 102]. For example, in the study by Nakata et al. (2018) [25], both humans and macaques detected faces of their own species faster compared to those of other species (measured through time to fixation in macaques and time to keypad response in humans). The authors of the study proposed that this increased efficiency in recognizing faces of one's own species is probably due to perceptual narrowing which happens in early life [103–105], and further experience-dependent maturation in later years, similar to what happens in the development of language in multilingual environments.

Further, studies in humans have shown that the accumulation of proper visual experience is crucial also for the development of holistic processing tendency, which persists even into adulthood [106–109]. This developmental process is paralleled by the maturation of higher visual areas in the brain, particularly those crucial for facial recognition [110–116]. Could there be simultaneous experience-dependent developmental changes in the eye movement patterns of humans that facilitate holistic face processing? By utilizing a combination of brain imaging and eye-tracking methods on both children and adults, the study by Gomez et al. [117] has provided support for this. Their results indicate that the spatial processing properties, particularly the population receptive field of face areas in children's brains continue to evolve into adulthood, displaying enlargement and a shift toward a more foveal orientation. Further, the changes in the spatial processing properties of face areas coincided with corresponding alterations in eye movement patterns. Compared to adults, whose first fixations were concentrated around the center of the nose, children exhibited less centralized first fixations. Following this line of reasoning, it is plausible that the less optimal holistic processing strategy observed in dogs when viewing human faces could be attributed to their relatively limited experience with human faces compared to human adult subjects in our experiment, despite that the dog subjects in our experiment encounter human individuals on a daily basis.

Another contributing factor to the less distinct AOI preference observed in dogs might be the inherent variability in eye movement patterns among individuals. This variability is well-documented in humans, with studies consistently showing systematic variation in the vertical locations of the first fixations across individuals [34, 39, 118, 119]. This variability is closely linked to individual differences in foveal structure and function [34, 37, 120–122]. Considering this, the observed first fixation patterns of dogs having greater variability is perhaps not so surprising. While the foundational structure and embryonic developmental plan of the retina are universal across vertebrates [123, 124], post-birth maturation and refinement, extending to higher visual areas, vary across species affected by species-specific evolutionary pressures. Furthermore, dogs, the most diverse species on Earth due to human-driven artificial selection [125, 126] exhibit many traits inherently linked to each breed, and some of them such as skull shape and eye laterality [127], retinal configuration [128], cerebral organization [129], and the distribution pattern of retinal ganglion cells [130] are directly linked to their vision. Hence, dogs exhibit a greater potential not only for more pronounced variations in fundamental aspects of eye movements compared to humans, such as the characteristics of saccades and fixations [65, 66], but also for their fixation patterns on faces. This could make it overall more challenging to discern distinct fixation preferences for areas of interest (AOIs) in the data of dogs compared to humans.

Further, the less effective holistic processing strategy seen in dogs when observing human faces, compared to faces of their own species, may be influenced by their reliance on a wider array of bodily gestures and sensory modalities during social interaction than humans. While humans also use bodily gestures to express emotions, the face is their primary medium for visually and verbally conveying subtle changes in emotional states [131]. More complex facial musculature and diverse muscle usage in humans than dogs support this [132]. On the other hand, dogs habitually use a broader range of their body, such as changes in posture and tail movement, for communication purposes compared to humans [131, 133]. Consequently, dogs might prioritize other bodily parts over the head or face when processing social cues. Correia-Caeiro et al. (2021)'s study involving dogs and body stimuli showed that dogs tended to focus more on the body rather than the head of human and dog figures [134]. Furthermore, as a species with highly developed olfactory senses, dogs might rely on non-visual communication cues, such as scent, during social interaction much more than humans do [135]. This varied means of emotional expression and the propensity for multisensory processing, differing from humans, may further lead to a reduced emphasis on visual information from faces in dogs and consequently less optimized fixation pattern especially on human faces.

## Conclusion

Our results support that humans demonstrate a pronounced holistic processing tendency when viewing human and dog faces, evident not only in their first fixation but also in subsequent fixations observed over seven seconds. Unlike humans, dogs did not exhibit a similar holistic processing tendency observed in their first fixations, but the results on the relative total fixation duration indicate that dogs might exhibit such tendency when they view the faces of their own species. Nevertheless, it is crucial to note that the accuracy and the amount of the data for dogs might limit our ability to draw firm conclusions.

Moving forward, studies should consider more optimal designation of AOIs, incorporating more ecological stimuli, such as live stimuli that better align with dogs' natural visual processing [18, 29, 136, 137], and employing more objective analysis techniques, such as a by-pixel test of different statistical pixel maps of fixations, which reveals statistically significant differences in pixels [30, 138]. Also, using additional tracking data as used in [64] on top of the

behavioral training for stimulus viewing and calibration would provide higher quality dog eye-tracking data. The combination of these approaches has the potential to offer a more profound understanding of dogs' face perception strategies. It can also illuminate the dynamics of inter-species communication between humans and dogs, and contribute to a comparative phylogenetic analysis of face-viewing behavior in animals. This, in turn, can shed light on the broader implications of human interactions with different species [19].

## Acknowledgments

We are grateful to all human participants and dog owners for their participation and tremendous support. We thank Catarina Espanca Bacelar, Sabrina Karl, and Celine Delay for their hard work in training dogs, Marlies Dolezal and Tamás Faragó for their advice on data analysis of pilot experiments, and Peter Füreder, Wolfgang Berger, and Karin Bayer and Jenny Bentlage for their excellent IT, technical, and administrative support, respectively. The authors would like to express their appreciation to the Cross Validated group of the Stack exchange community for valuable information on statistics and related computation using R.

## Author Contributions

**Conceptualization:** Soon Young Park, Ludwig Huber, Zsófia Virányi.

**Data curation:** Soon Young Park, Diederick C. Niehorster.

**Formal analysis:** Soon Young Park, Diederick C. Niehorster.

**Funding acquisition:** Ludwig Huber, Zsófia Virányi.

**Investigation:** Soon Young Park.

**Methodology:** Soon Young Park, Ludwig Huber, Zsófia Virányi.

**Project administration:** Soon Young Park, Zsófia Virányi.

**Resources:** Ludwig Huber, Zsófia Virányi.

**Software:** Diederick C. Niehorster.

**Supervision:** Diederick C. Niehorster, Ludwig Huber, Zsófia Virányi.

**Validation:** Soon Young Park, Diederick C. Niehorster.

**Visualization:** Soon Young Park.

**Writing – original draft:** Soon Young Park.

**Writing – review & editing:** Soon Young Park, Diederick C. Niehorster, Ludwig Huber, Zsófia Virányi.

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
