## [Decision Letter · Decision Letter 0]

28 Nov 2024

PONE-D-24-35898Examining Holistic Processing Strategies in Dogs and Humans Through Gaze BehaviorPLOS ONE

Dear Dr. Park,

Thank you for submitting your manuscript to PLOS ONE. After careful consideration, we feel that it has merit but does not fully meet PLOS ONE’s publication criteria as it currently stands. Therefore, we invite you to submit a revised version of the manuscript that addresses the points raised during the review process.

We look forward to receiving your revised manuscript.

Kind regards,

Stanisław Jacek Wroński, M.D., Ph.D, FEBU

Academic Editor

PLOS ONE

Additional Editor Comments:

Dear Authors,

after careful consideration of the manuscript and reviewer's opinion, I conclude that in its present form the article needs minor revision.

The comments mainly concern the technical aspects , that is, the quality and layout of the illustrations.

This point needs necessary improvement.

With compliments

Stanisław Wroński

Academic Editor

Reviewers' comments:

Reviewer's Responses to Questions

**Comments to the Author**

1. Is the manuscript technically sound, and do the data support the conclusions?

Reviewer #1: Yes

2. Has the statistical analysis been performed appropriately and rigorously? 

Reviewer #1: Yes

3. Have the authors made all data underlying the findings in their manuscript fully available?

Reviewer #1: Yes

4. Is the manuscript presented in an intelligible fashion and written in standard English?

Reviewer #1: Yes

5. Review Comments to the Author

Reviewer #1: The present study examines the gaze patterns of both human and dog participants. This is a meaningful topic, and the manuscript is well-written. However, there are two minor issues that need to be addressed before it can be accepted for publication.

1. The images in the "else upper half" column and those in the "else lower half" column appear to be misplaced and should be switched.

2. The figures are all quite unclear. There is a need for higher resolution figures.

6. PLOS authors have the option to publish the peer review history of their article (what does this mean?). If published, this will include your full peer review and any attached files.

Reviewer #1: No

---

## [Author Response · Author response to Decision Letter 0]

2 Dec 2024

Dear Reviewer #1,

Thank you for your thorough review and kind feedback on the manuscript. Especially, we appreciate you pointing out the critical error in Figure 3. We have addressed the issues you raised as follows:

Figure Placement: The images in the "else upper half" and "else lower half" columns of Figure 3 have been corrected and are now properly positioned.

Figure Resolution: All figures have been replaced with higher-resolution versions for improved clarity. 

Thank you again for taking the time to review our manuscript. 

On behalf of all authors,

Sincerely,

Soon Young Park

---

## [Decision Letter · Decision Letter 1]

29 Dec 2024

Examining Holistic Processing Strategies in Dogs and Humans Through Gaze Behavior

PONE-D-24-35898R1

Dear Dr. Soon Young Park

We’re pleased to inform you that your manuscript has been judged scientifically suitable for publication and will be formally accepted for publication once it meets all outstanding technical requirements.

Kind regards,

Stanisław Jacek Wroński, M.D., Ph.D, FEBU

Academic Editor

PLOS ONE

Additional Editor Comments (optional):

Dear Authors,

after after careful consideration of the revised version of the article entitled: "Examining Holistic Processing Strategies in Dogs and Humans Through Gaze Behavior" (PONE-D-24-35898R1) I find that the submitted article in the revised version is suitable for the journal PLOS ONE.

With compliments

Stanisław Wroński

PLOS ONE

Academic Editor

---

## [Editor Report · Acceptance letter]

17 Jan 2025

PONE-D-24-35898R1 

PLOS ONE

Dear Dr. Park, 

I'm pleased to inform you that your manuscript has been deemed suitable for publication in PLOS ONE. Congratulations! Your manuscript is now being handed over to our production team.

Kind regards, 

on behalf of

Dr. Stanisław Jacek Wroński 

Academic Editor

PLOS ONE